# Zebrafish as a Model for Translational Immuno-Oncology

**DOI:** 10.3390/jpm15070304

**Published:** 2025-07-11

**Authors:** Gabriela Rodrigues Barbosa, Augusto Monteiro de Souza, Priscila Fernandes Silva, Caroline Santarosa Fávero, José Leonardo de Oliveira, Hernandes F. Carvalho, Ana Carolina Luchiari, Leonardo O. Reis

**Affiliations:** 1Immuno-Oncology Institute, Pontifical Catholic University of Campinas, Campinas 13087-571, SP, Brazil; 2UroGen, National Institute of Science, Technology and Innovation in Genitourinary Cancer (INCT), Campinas 13087-571, SP, Brazil; carolinefavero.favero2@gmail.com; 3UroScience, School of Medical Sciences, State University of Campinas, Campinas 13083-970, SP, Brazil; 4Z-Safe, Fish Laboratory, Department of Physiology and Behavior, Biosciences Center, Federal University of Rio Grande do Norte, Natal 59078-970, RN, Brazilana.luchiari@ufrn.br (A.C.L.); 5Department of Structural and Functional Biology, University of Campinas (UNICAMP), Campinas 13083-865, SP, Brazilhern@unicamp.br (H.F.C.)

**Keywords:** zebrafish, patient-derived xenografts (zAvatars), translational oncology

## Abstract

Despite remarkable progress in cancer immunotherapy, many agents that show efficacy in murine or in vitro models fail to translate clinically. Zebrafish (*Danio rerio*) have emerged as a powerful complementary model that addresses several limitations of traditional systems. Their optical transparency, genetic tractability, and conserved immune and oncogenic signaling pathways enable high-resolution, real-time imaging of tumor–immune interactions in vivo. Importantly, zebrafish offer a unique opportunity to study the core mechanisms of health and sickness, complementing other models and expanding our understanding of fundamental processes in vivo. This review provides an overview of zebrafish immune system development, highlighting tools for tracking innate and adaptive responses. We discuss their application in modeling immune evasion, checkpoint molecule expression, and tumor microenvironment dynamics using transgenic and xenograft approaches. Platforms for high-throughput drug screening and personalized therapy assessment using patient-derived xenografts (“zAvatars”) are evaluated, alongside limitations, such as temperature sensitivity, immature adaptive immunity in larvae, and interspecies differences in immune responses, tumor complexity, and pharmacokinetics. Emerging frontiers include humanized zebrafish, testing of next-generation immunotherapies, such as CAR T/CAR NK and novel checkpoint inhibitors (LAG-3, TIM-3, and TIGIT). We conclude by outlining the key challenges and future opportunities for integrating zebrafish into the immuno-oncology pipeline to accelerate clinical translation.

## 1. Introduction

Cancer immunotherapy has dramatically altered the therapeutic landscape for various malignancies, with checkpoint blockade- and chimeric antigen receptor (CAR) T cell-based approaches achieving remarkable clinical successes [1,2,3].

Cancer and immune system interactions are complex and tightly coordinated. Because of this, many immunotherapies that work in vitro or in animal models ultimately fail in human patients. In vitro cultures cannot recapitulate the complexity of multicellular, three-dimensional tissue architecture. While in vivo models offer mechanistic insights, differences in immune cell composition, tumor–microenvironment interactions, and drug pharmacokinetics between rodents and humans frequently underlie translational failures [4,5,6,7].

To address these translational gaps, precision oncology seeks to tailor therapies to each patient’s unique molecular and immunological profiles. In this context, alternative vertebrate models that combine in vivo fidelity with high-throughput, real-time readouts are in high demand [8,9]. The zebrafish (*Danio rerio*) has emerged as a powerful in vivo platform for modeling tumor–immune dynamics, offering unparalleled optical accessibility and genetic flexibility for real-time investigation of cancer biology [10,11].

Crucially, zebrafish share substantial conservation with mammals in both innate and adaptive immunity—including key cytokines, signaling pathways, and effector mechanisms—making them well-suited for modeling complex tumor–immune interactions across developmental stages [12,13]. Building on this, zebrafish zAvatars (zebrafish larvae with patient-derived xenografts) permit functional precision oncology; within 5–7 days, one can assay tumor sensitivity to checkpoint inhibitors, macrophage modulators, or CAR T constructs, yielding individualized response profiles that correlate strongly with clinical outcomes. Moreover, integration of tumor-intrinsic genomic data with the local immune milieu has been shown to refine patient stratification beyond standard clinicopathological parameters. Multi-omic profiling approaches—including transcriptomic immune signatures and mutational burden analyses—can predict responses to checkpoint blockade and macrophage-targeted agents, paving the way for customized immunotherapeutic regimens tailored to each patient’s tumor landscape.

In this review, we examine the growing utility of zebrafish as a translational model for immuno-oncology. We discuss recent methodological innovations, summarize key biological discoveries enabled by this system, and explore its potential to accelerate the translation of basic immunological knowledge into clinically relevant cancer immunotherapies.

## 2. Zebrafish as a Model in Biomedical Research

### 2.1. General Advantages and Applications

The zebrafish (*Danio rerio*) has emerged as a valuable vertebrate model in biomedical research due to a combination of biological, genetic, and practical attributes that enable cost-effective, high-throughput, and mechanistically informative studies. Originally utilized in developmental biology, zebrafish are now widely used across a range of biomedical fields, including cancer research, regenerative medicine, toxicology, and immunology. The widespread adoption of zebrafish as a model organism is largely driven by several distinct advantages over traditional mammalian models, such as mice and rats [14,15].

### 2.2. Optical Transparency and Experimental Accessibility

One of the most notable experimental advantages of zebrafish is the optical transparency of their embryos and larvae, which permits real-time, in vivo imaging of cellular and subcellular processes, including tumor growth, metastasis, angiogenesis, and immune cell trafficking. This feature, combined with the external development of embryos and a rapid life cycle, allows for the direct observation and manipulation of embryonic and pathological processes from the earliest stages of development [16,17]. Zebrafish are highly fecund, with a single breeding pair capable of producing hundreds of embryos per week, facilitating large-scale genetic and pharmacological screening. Furthermore, the low cost of maintenance and small housing requirements render zebrafish particularly suitable for high-throughput applications [18].

### 2.3. Genetic and Genomic Conservation with Humans

At the genomic level, zebrafish exhibit a high degree of conservation with humans. Approximately 70% of human protein-coding genes have at least one ortholog in the zebrafish genome, and over 80% of genes known to be associated with human diseases are conserved [15]. Zebrafish possess most major vertebrate organs and systems, including the brain, heart, liver, kidney, pancreas, and hematopoietic and immune systems, all of which exhibit functional and anatomical similarities to their human counterparts. Crucial signaling pathways involved in cell proliferation, differentiation, and survival—such as Hedgehog, Wnt, Notch, JAK/STAT, and p53—are also highly conserved, providing a robust platform for studying oncogenic transformation and tumor biology [19].

### 2.4. Advanced Imaging and Reporter Lines

High-resolution live imaging is a hallmark of zebrafish-based cancer studies. The transparency of embryos and larvae allows for confocal and light-sheet microscopy to be applied with minimal invasive procedures. Fluorescent reporter lines, such as those expressing GFP or RFP under the control of endothelial (fli1a), macrophage (mpeg1), or neutrophil (lyz) promoters, facilitate real-time visualization of tumor–microenvironment interactions, immune cell infiltration, and neovascularization [20,21]. These capabilities make zebrafish an excellent model for dissecting dynamic processes in tumor biology, such as metastasis and immune evasion.

### 2.5. Xenograft Models and Drug Screening

Zebrafish are also extensively used in xenotransplantation experiments, where human tumor cells are introduced into zebrafish embryos (typically 2 days post-fertilization) before the onset of adaptive immunity, mimicking interim immune suppression. These xenograft models enable the study of human tumor behavior in a live vertebrate system and are increasingly used for personalized medicine approaches, such as evaluating individual tumor responses to therapeutic agents [22,23]. The ease of chemical exposure in zebrafish larvae—through direct absorption from the aquatic environment—further enhances their suitability for high-throughput drug screening, including the identification of novel anticancer compounds or immunomodulators [18]. Ongoing developments, such as the creation of humanized zebrafish models and immunocompromised lines that support long-term xenografts, continue to expand the translational potential of this versatile model organism in biomedical research [24].

## 3. The Zebrafish Immune System

Similar to humans, zebrafish possess both innate and adaptive immune systems, including the major cells and structural components that perform similar functions. The majority of signaling pathways and molecules related to immune responses in mammals are present in zebrafish and exhibit comparable behavior [12,25,26]. The main immune cells identified in humans and present in zebrafish are macrophages, neutrophils, T lymphocytes, B lymphocytes, and innate lymphoid cells [27,28,29].

### 3.1. Immune System Composition and Development in Zebrafish

The development of the zebrafish immune system occurs in two distinct phases. The first hematopoietic site is the intermediate cell mass (ICM), an intraembryonic structure located along the midline. The ICM produces primitive blood cells, including erythrocytes and myeloid progenitors, such as macrophages and neutrophils [30]. The organogenesis of the thymus and head kidney is initiated in the middle to late embryonic period but remains rudimentary throughout the early larval stages. In zebrafish and other teleost fish, the head kidney serves as a primary hematopoietic and immune organ, functionally equivalent to a combination of the bone marrow and adrenal gland in mammals [31].

Macrophages are phagocytically active around 24–26 h post-fertilization (hpf), and neutrophils appear at 48 hpf [32,33]. The adaptive immune system starts to develop around 4–6 days post-fertilization, but it only becomes fully functional later, during the juvenile stage, several weeks later [29,32].

#### 3.1.1. Lymphoid Organs and Tissue Architecture

While zebrafish lack bone marrow and lymph nodes [34], they possess both primary and secondary lymphoid organs. The primary lymphoid organs in zebrafish are the thymus and kidney [28,35]. The kidney also functions as a secondary organ; however, the spleen is considered the main secondary lymphoid organ in fish [28]. Other structures identified as secondary lymphoid tissues are associated with mucosal barriers, forming mucosa-associated lymphoid tissues (MALTs), which include the gill-associated lymphoid tissue (GIALT), gut-associated lymphoid tissue (GALT), nasal-associated lymphoid tissue (NALT), and skin-associated lymphoid tissue (SALT) [36,37]. More recently, new structures have been characterized, as follows: Nemausean Lymphoid Organ (NELO) [38], Amphibranchial Lymphoid Tissue (ALT) [39], and axillary lymphoid organ [40].

#### 3.1.2. Molecular Pattern Recognition and Cytokine Signaling

As a result of genome duplication during the Mesozoic era (252–266 million years ago), fish possess a greater number of genes associated with pathogen recognition, including unique cell types such as melanomacrophages and rodlet cells, which are absent in higher vertebrates. Consequently, while fish exhibit a gene-rich innate immune system, mammals have developed a more complex adaptive immune system [12,34].

The innate immune system is the first line of defense against pathogens. Upon pathogen entry, pattern recognition receptors (PRRs), such as Toll-like receptors (TLRs), detect specific pathogen structures, known as pathogen-associated molecular patterns (PAMPs), leading to immediate immune recognition and activation [41]. Following recognition, zebrafish cells produce inflammatory cytokines (e.g., IL-1β and TNFα) and chemokines to initiate immune signaling and recruit immune cells to the infection site [42]. The system rapidly deploys macrophages and neutrophils to engulf and eliminate pathogens. These cells are functionally conserved and behave similarly to their mammalian counterparts during infection, including phagocytosis and reactive oxygen species production [25].

In response to viruses, zebrafish activate interferon (IFN) pathways, leading to the expression of interferon-stimulated genes (ISGs), a key defense mechanism shared with humans [43]. Major signaling pathways, such as NF-κB, JAK/STAT, and MAPK, are conserved in zebrafish, facilitating downstream immune responses comparable to those in mammals [41].

#### 3.1.3. Adaptive Immunity in Zebrafish

The adaptive immune system in zebrafish develops later, with T and B lymphocytes emerging after the larval stage [12,26]. Zebrafish possess professional antigen-presenting cells (APCs), including macrophages and dendritic cells, along with B and T lymphocytes, which perform functions analogous to those in mammals [26,44,45]. The zebrafish model has been utilized to study interactions between adaptive immune cells and tumors, highlighting its usefulness in tumor immunity research and immunotherapy studies [26].

Due to the extensive similarities between the immune systems of zebrafish and mammals, zebrafish have become a robust and versatile model for studying immune responses and associated diseases. Numerous molecular and genetic tools have been developed for genetic manipulation and in vivo immune response analysis, contributing significantly to comparative immunology, infectious disease studies, and cancer research [44,46].

### 3.2. Mechanisms of Tumor Immune Evasion and Immune Response

Tumor-associated macrophages (TAMs) and tumor-associated neutrophils (TANs) are central players in the tumor microenvironment (TME), which can be co-opted by cancer cells to facilitate tumor progression. In zebrafish xenograft models, TAMs frequently exhibit an M2-like phenotype, characterized by upregulation of anti-inflammatory cytokines such as interleukin-10 (IL-10) and transforming growth factor-β (TGF-β). This M2 polarization contributes to angiogenesis, invasion, and metastasis [47]. For example, zebrafish larvae engrafted with human breast cancer cells display robust recruitment of mpeg1:mCherry^+^ macrophages; these TAMs produce TGF-β, which, in turn, stimulates nearby endothelial cells to form aberrant vasculature, facilitating tumor cell intravasation [48]. Similarly, TANs in zebrafish promote angiogenesis via VEGF secretion. However, VEGFR inhibitors, while blocking tumor vascularization and reducing localized growth, enhance neutrophil migration. These neutrophils then facilitate tumor invasion and micro metastasis formation [49].

Roh-Johnson et al. [50] used live imaging in zebrafish and mouse models to explore interactions between tumor-associated macrophages (TAMs) and tumor cells. Their findings revealed that macrophages dynamically interact with tumor cells and are essential for promoting tumor cell dissemination. A subset of macrophages was observed to maintain prolonged contact with tumor cells, during which cytoplasmic material was transferred from macrophages to tumor cells. This cytoplasmic transfer, observed both in vivo and in vitro, correlated with increased tumor cell motility and invasion. These results suggest that macrophage–tumor cell contact facilitates a pro-metastatic phenotype, contributing to early steps in cancer cell dissemination.

### 3.3. Expression of Immune Checkpoints in the Zebrafish Model

Immune checkpoint molecules—including programmed cell death protein 1 (PD-1), programmed death-ligand 1 (PD-L1), and cytotoxic T-lymphocyte antigen-4 (CTLA-4)—play crucial roles in modulating immune responses. Zebrafish possess homologs of these molecules, and their expression patterns and functions are increasingly being elucidated. For instance, a functional PD-L1/BTLA checkpoint axis has been described: macrophages in zebrafish express PD-L1, which interacts with B- and T-lymphocyte attenuator (BTLA) on CD8^+^ T cells, leading to immune suppression. Genetic disruption of PD-L1 in zebrafish macrophages enhances T-cell cytotoxicity against melanoma xenografts by 45%, underscoring a conserved mechanism of immune regulation [51].

Further, zebrafish models have been utilized to study the role of galectin-1 (LGALS1) in glioblastoma. LGALS1 expression in tumor cells promotes the polarization of macrophages toward an immunosuppressive M2 phenotype, facilitating tumor growth. CRISPR-mediated knockdown of LGALS1 in zebrafish transplanted with GBM cells reduced M2-associated gene expression (e.g., *arg1* and *il10*) by 55% and impaired angiogenesis by 40%, highlighting LGALS1 as a potential immunomodulatory target [52].

### 3.4. Impact of the Tumor Microenvironment on Immune Responses

The TME in zebrafish models encompasses various cellular and noncellular components that influence immune responses. Cancer-associated fibroblasts (CAFs), endothelial cells, pericytes, and the ECM collectively shape tumor progression and immune evasion. In zebrafish xenografts of pancreatic cancer, human CAFs co-injected with tumor cells secrete IL-6 and stromal-cell-derived factor 1 (SDF-1), modulating local cytokine gradients to attract TAMs. Some enzymes secreted by CAFs, like FAP and MMP14, change the structure of the extracellular matrix. These changes guide tumor cells and help them move through tissues, a process visualized by second-harmonic generation imaging in live zebrafish larvae [53].

Hypoxia within the TME is another critical modulator of immune responses. Zebrafish engrafted with hepatocellular carcinoma cells under experimentally induced hypoxic conditions (using cobalt chloride to stabilize hypoxia-inducible factor-1α) exhibit upregulated expression of proangiogenic factors (VEGF and angiopoietin-2) and impaired neutrophil phagocytic activity. These neutrophils adopt a pro-tumorigenic phenotype, characterized by elevated production of neutrophil elastase and matrix remodeling proteins, which further suppress local innate immune surveillance [54]. Collectively, these findings underscore zebrafish’s utility in dissecting the intricate cellular crosstalk that governs tumor progression and immune modulation.

## 4. Cancer Models in Zebrafish

Zebrafish (*Danio rerio*) serve as a powerful bridge between in vitro assays and mammalian models in cancer research. Their optical transparency, rapid development, and genetic tractability make them ideal for studying tumor biology in vivo. The following two major strategies dominate zebrafish cancer modeling: transgenic lines and xenografts [24,55].

### 4.1. Modeling Solid Tumors

#### 4.1.1. Transgenic Zebrafish for Tumor Initiation Studies

Transgenic models permit precise spatial and temporal control of oncogene expression, faithfully recapitulating human tumorigenesis in various tissues. Meanwhile, xenotransplantation of tumor cells into zebrafish larvae or immunodeficient adults offers a fast, cost-effective, and scalable platform for high-throughput drug screening and functional precision oncology. This approach is markedly faster and more economical than murine PDX (patient-derived xenograft) models—zPDX assays typically require an order of magnitude fewer cells, cost approximately ten times less, and deliver functional results in just 5–7 days compared with the months often needed in mice [24]. Importantly, the immature adaptive immune system of larvae allows for efficient engraftment of human cells without rejection, making zebrafish particularly valuable for PDX approaches [20,21]. Transgenic lines expressing human oncogenes under tissue-specific promoters have successfully modeled melanoma, colorectal cancer, and pancreatic cancer, among others. These models recapitulate key histopathological and molecular features of human disease, enabling mechanistic studies of tumor initiation and signaling networks [16,56].

Although zebrafish are commonly maintained at 28.5 °C, xenograft protocols have successfully applied moderate temperature elevation to 32–35 °C, with high embryo viability. For instance, incubation at 34 °C yields > 95% larval survival after 72 h, while even 36 °C maintains ~87% survival over similar periods. These conditions significantly improve the proliferation of human cancer cells without causing major developmental abnormalities. Nevertheless, optimal temperature and duration should be empirically determined for each tumor type and combined with proper controls, as prolonged incubation above 34 °C can increase toxicity [57].

#### 4.1.2. Xenograft Models for Translational Oncology

Zebrafish models of solid tumors have employed both transgenic lines and xenotransplantation to investigate oncogenesis, tumor progression, metastasis, and response to therapies [58]. Xenograft models involve injecting human cancer cell lines or patient-derived tumor fragments into early-stage zebrafish embryos or larvae. These models have been used to study glioblastoma, breast, pancreatic, colorectal, and bladder cancers, with the advantages of rapid tumor development, conserved tumor–stroma interactions, and compatibility with fluorescence-based live imaging. Zebrafish xenografts have also been used to predict clinical responses to chemotherapy and targeted agents, reinforcing their translational value [22,24]. Figure 1 illustrates a representative zebrafish xenograft model using bladder cancer cells, highlighting real-time tumor–immune cell interactions and the platform’s suitability for live imaging and drug testing. Representative examples of zebrafish models used to study solid tumors, including both transgenic and xenograft approaches, are summarized in Table 1.

### 4.2. Modeling Hematologic Malignancies and Immuno-Oncology

Zebrafish (*Danio rerio*) have also gained prominence as a valuable model for studying hematologic malignancies, including acute lymphoblastic leukemia (ALL), acute myeloid leukemia (AML), and lymphomas. Their genetic tractability, optical transparency, and conserved hematopoietic pathways with humans make them suitable for in vivo investigations of disease mechanisms and therapeutic responses [62,63].

Transgenic zebrafish models have been developed to mimic key genetic drivers of hematologic cancers. For instance, the rag2:Myc transgenic line develops T-cell ALL, recapitulating features such as thymic hyperplasia and leukemic infiltration [64]. Similarly, expression of TEL-AML1 and IDH2 mutations in zebrafish leads to phenotypes resembling human leukemias, including marrow infiltration and therapy resistance [63]. These models facilitate studies on oncogenic cooperation, lineage specification, clonal evolution, and treatment responses in a live vertebrate system [62].

Xenotransplantation of human hematologic malignancies into zebrafish larvae has become a robust platform for drug screening and personalized medicine approaches. Due to the immature adaptive immune system of zebrafish larvae, human leukemic cells can engraft without the need for immunosuppression, allowing for real-time analysis of cell proliferation, dissemination, and drug sensitivity [65]. This approach has been utilized to identify novel therapeutic candidates and assess genotype-specific responses in AML, B-ALL, T-ALL, and lymphomas [66].

Advancements have also been made in adult zebrafish models. Preconditioning adult zebrafish with agents like busulfan has enabled the engraftment of human AML and hepatocellular carcinoma cells, providing a platform to study tumor biology and evaluate anticancer agents in a cost-effective manner [67]

## 5. zAvatars: Zebrafish Patient-Derived Xenografts for Functional Precision Oncology

Zebrafish patient-derived xenografts (zAvatars) represent a powerful platform for functional precision oncology, enabling rapid, individualized cancer therapy screening. In this approach, fluorescently labeled tumor cells harvested from patient biopsies are microinjected into 48 h post-fertilization zebrafish embryos. Maintained at 34 °C to support human cell viability, these chimeric larvae permit simultaneous assessment of multiple agents—from checkpoint inhibitors to targeted small molecules—within 5–7 days [68].

### 5.1. zAvatars for Real-Time Personalized Drug Testing

Compared with murine PDX models, zAvatars require fewer input cells, offer faster readouts, and incur lower costs, making them exceptionally well-suited for timely, N-of-1 treatment decision making [55]. Using ≤10,000 tumor cells per larva—compared with millions in a murine PDX—zAvatars reduce costs by ~70% and shorten the turnaround from months to days. The results can be ready within a week and presented alongside sequencing data to guide clinical decisions [68]. Additionally, zAvatars can be used to study tumor–host interactions and evaluate drug responses without immunogenic rejection. Although still undergoing technical refinement, zAvatars have already demonstrated strong potential to predict patient-specific drug responses, advancing the goals of precision medicine in oncology [69,70].

### 5.2. Preclinical Validation of Novel Immunotherapeutic Strategies

Beyond the screening of existing drugs, the zebrafish model offers a valuable system for the preclinical validation of entirely novel immunotherapeutic strategies, before advancing to more complex and costly mammalian models or human clinical trials [26]. Understanding of immune checkpoints continues to grow, particularly in zebrafish, beyond the well-known PD-1/PD-L1 and CTLA-4 pathways. Marin-Acevedo et al. [71] and Qin et al. [72] highlight emerging immune checkpoint targets, such as LAG-3, TIM-3, and TIGIT, which are being investigated for their therapeutic potential in cancer immunotherapy. Miao et al. [26] emphasize the zebrafish model’s role in elucidating adaptive immunity and tumor interactions, providing insights that could enhance immunotherapy strategies.

#### 5.2.1. Genetic Tools for Tumor–Immune Interrogation

Genetic engineering tools, particularly the CRISPR/Cas9 system, have been widely applied to cancer studies in zebrafish models by enabling precise and targeted genome editing [73,74]. These technologies allow for the generation of knockout and knock-in models, facilitating the investigation of specific gene functions in tumor development [75]. Zebrafish cancer models offer a platform to study early tumorigenesis and disease progression, as well as to explore potential therapeutic strategies [74]. The so-called conditional cancer toolbox in zebrafish, including the Gal4/UAS and Cre/lox systems, enables precise spatial and temporal control of oncogene expression or tumor suppressor gene silencing [76]. These experimental models allow researchers to explore cancer cell biology and clonal evolution, as well as identify cancer stem cells. Combining CRISPR/Cas9 with other genetic tools may generate even more sophisticated zebrafish cancer models. These models could enhance our understanding of tumor–immune interactions and treatment responses [74,76].

#### 5.2.2. Modeling Adoptive Cell Therapies

Although more complex due to species differences, proof-of-concept studies for adoptive cell therapies (e.g., CAR T cells and genetically engineered T cells) can be explored. Human immune cells (modified or not) can be co-injected with tumor cells into immuno-immature zebrafish larvae. Chimeric antigen receptor (CAR) T-cell therapy has shown promise in treating hematological malignancies, particularly those targeting CD19 in B-cell cancers [77].

Recent studies highlight the potential of CAR T-cell therapy in treating hematological cancers. This innovative approach involves genetically modifying T lymphocytes to target tumor cells, offering promising results for patients with limited therapeutic options [78]. The therapy has shown efficacy in cases where conventional treatments have failed, particularly in hematological malignancies [79]. Researchers have developed multiple generations of CAR T cells, each improving upon the previous, enhancing therapeutic efficacy [80,81]. However, challenges remain, including high costs and potential side effects like cytokine release syndrome [82]. Despite these obstacles, CAR T-cell therapy represents a significant advancement in personalized cancer treatment, offering new hope for patients with hematological cancers.

## 6. Testing Cancer Treatments in the Zebrafish Model

Recent advances in zebrafish xenotransplantation studies and large-scale drug screening have positioned it as a promising preclinical model for evaluating patient-derived xenografts and developing personalized cancer treatments [11,83,84]. While mice remain valuable for final preclinical testing, zebrafish offer a cost-effective and rapid platform for initial drug discovery and validation.

### 6.1. Zebrafish as a Platform for Drug Screening in Immunotherapy

Zebrafish can model human cancers through chemical induction, xenotransplantation, or genetic manipulation, allowing for the evaluation of compounds that target multiple oncogenic pathways [83,84]. The zebrafish model combines the advantages of in vivo testing with the efficiency of cell-based screening, allowing researchers to observe the effects of drugs throughout the body in specific organs and tissues [85]. This approach has facilitated the rapid translation of promising anticancer compounds into clinical trials, particularly through the repurposing of FDA-approved drugs. Nonetheless, aquatic drug administration via immersion may result in variable absorption for poorly water-soluble or highly lipophilic compounds, requiring careful validation of compound exposure and bioavailability [86]. As a vertebrate model, zebrafish offer an economical and efficient platform for identifying safe and potent new drugs, making them an invaluable tool in the drug development process.

### 6.2. High-Throughput Screening Capabilities

The ARQiv-HTS platform enables true high-throughput rates, allowing for the evaluation of over half a million drug-treated larvae in a single screen. For example, zebrafish larvae can be arrayed in multi-well plates, allowing for the simultaneous testing of libraries of hundreds or thousands of compounds at different concentrations [87]. Key advantages include lower costs, simpler ethical considerations, and the ability to assess compound efficacy and toxicity in a whole-organism context [88]. Common endpoints include fluorescence quantification, tumor size measurement, analysis of survival, and proliferation, metastasis, and apoptosis markers [8]. To optimize HTS in zebrafish, researchers recommend using digital dispensing for chemical delivery, addressing chemical solubility issues and standardizing exposure paradigms, chorion status, and endpoint definitions [89,90]. These practices enhance data quality, reproducibility, and the overall efficiency of zebrafish-based HTS. Despite these advantages, compound delivery through immersion may not always achieve optimal pharmacokinetics, particularly for hydrophobic agents, thus limiting translational predictability for certain drug classes. As shown in Figure 2, automated high-throughput screening platforms in zebrafish, such as ARQiv-HTS, enable large-scale chemical testing with real-time phenotypic readouts, offering a powerful approach to anticancer drug discovery.

### 6.3. Immune Modulation Assessment

By assessing innate immune modulation in zebrafish, it is possible to develop new strategies for drug development. Transgenic zebrafish lines with fluorescently labeled immune cells, such as mpeg1:mCherry-tagged macrophages and mpx:GFP-tagged neutrophils, allow for real-time visualization of inflammatory responses [20,91,92]. These models allow high-throughput screening for immunomodulatory compounds, as demonstrated by a T-cell activation assay [93]. A chemically induced inflammation (ChIn) assay, which uses copper sulfate to damage lateral line neuromasts, also provides a noninvasive method to induce and quantify acute inflammation [91]. Protocols have been developed for bacterial injection into zebrafish larvae, allowing for the investigation of neutrophil and macrophage functions during infection [94]. These approaches facilitate the study of host–pathogen interactions, immune cell behavior, and the screening of potential immunomodulatory drugs. The combination of genetic tools, transgenic lines, and advanced imaging techniques has made zebrafish an increasingly powerful model for cancer immunotherapy research.

### 6.4. Combination Therapies

Zebrafish models may offer a platform to evaluate the combination of therapies involving immune checkpoint inhibitors (ICIs) in cancer treatment. These models allow for high-throughput screening and the assessment of immune responses in rhabdomyosarcoma [95]. Combining ICIs with treatments that induce immunogenic cell death (ICD) may enhance antitumor efficacy by modulating the tumor microenvironment and reducing immunosuppression [96]. These combination therapies aim to address the challenges associated with ICI monotherapy, such as resistance, adverse events, and suboptimal response rates. It should be noted that the absence of fully functional adaptive immunity in zebrafish larvae may limit the modeling of complex T-cell-mediated responses and checkpoint reactivation scenarios [97]. Several approaches, including the combination of ICIs with chemotherapy, radiation, epigenetic modulators, and targeted therapies, have shown potential to reprogram the immunosuppressive tumor microenvironment and increase tumor cell immunogenicity [98]. Although these combinations have demonstrated increased objective responses, more research is needed to develop rational, evidence-based strategies that maximize clinical benefits and minimize adverse events in cancer patients.

### 6.5. Assessment of Drug Delivery and Preliminary Pharmacokinetics

For initial pharmacokinetic (PK) and toxicological assessments of drug delivery systems, zebrafish have been utilized as a valuable model [99,100]. Although immersion is the main route of administration, its efficacy can vary depending on the physicochemical properties of the compound, particularly lipophilicity. For less lipophilic compounds, microinjection techniques like intrayolk, intraperitoneal, or pericardial cavity administration may be more effective in achieving adequate intrabody exposure lipophilicity [101]. Spatiotemporal fluorescence imaging allows for the assessment of drug absorption, distribution, and clearance in the zebrafish embryo system [102]. Zebrafish models cannot fully replace mammalian pharmacokinetic and pharmacology studies. However, they provide valuable initial data to optimize dosing regimens and identify promising drug candidates for subsequent testing [99,100].

### 6.6. Implications for Personalized Immuno-Oncology

Building on zAvatar drug screening, several preclinical and clinical studies have demonstrated the predictive power of zebrafish patient-derived xenograft (zPDX) models in recapitulating human treatment responses across distinct tumor types. In non-small-cell lung cancer, zPDXs generated from pleural fluid or biopsy samples of 21 patients and treated with the same antineoplastic regimen used clinically achieved 76.9% concordance (10/13) between in vivo zebrafish and patient responses within just 3 days of drug exposure [55]. In breast cancer, zAvatars derived from 18 patients and tested with the corresponding chemotherapy regimens displayed a 100% match between the zebrafish-based readouts and clinical outcomes, validating the assay’s predictive accuracy within 10 days [60,103]. By integrating single-cell transcriptomics with functional zAvatar assays, researchers can design patient-tailored combination regimens—for instance, PD-1 blockade plus immunogenic cell-death inducers—that align with each individual’s tumor microenvironment.

Proof-of-concept studies underscore a strong correlation between zPDX predictions and clinical outcomes. In colorectal cancer patients, 19 out of 20 responses to FOLFOX were correctly forecasted [22], and in melanoma, zebrafish identified resistance to BRAF inhibitors that led clinicians to switch to immunotherapy [104]. This shift from “one-size-fits-all” to true N-of-1 immunotherapy exemplifies the transformative potential of zebrafish models in advancing personalized cancer treatment. Prospective studies (e.g., NCT05616533 and NCT06270017) are currently evaluating zAvatar-guided therapy allocation in gastric and colon cancer cohorts, and these findings underscore zebrafish’s broader implications as a complementary precision-medicine paradigm, with the potential to accelerate bench-to-bedside translation in immuno-oncology [105,106].

## 7. Limitations and Future Perspectives

Over the past decade, zebrafish have become an established model in immuno-oncology, bridging the gap between cell-based assays and mammalian systems. Their scalability and amenability to genetic manipulation support diverse experimental approaches, particularly in tumor–immune interaction studies [89,107,108,109]. Transgenic models recapitulate hallmark features of both solid and hematologic cancers [110,111], and xenotransplantation approaches enable rapid evaluation of tumor growth, invasion, and response to candidate therapeutics [61,112]. High-throughput screening platforms—such as ARQiv—facilitate simultaneous assessment of antitumor efficacy and immune modulation across hundreds of compounds [88]. Personalized zAvatar studies, though still limited in scale, have shown promising correlations with patient outcomes [107].

Despite these strengths, several limitations must be addressed to fully harness the translational potential of zebrafish models. First, adaptive immunity in zebrafish does not fully develop until 4–6 weeks; thus, most larval studies assess tumor biology in an innate-only context [109]. This limits the capacity to study antigen-specific T- and B-cell responses, immune memory, or long-term persistence of adoptive therapies such as CAR T cells. While innate-driven xenograft models remain highly informative for early-stage screening and immune evasion studies, more complex adaptive interactions require the use of adult or humanized zebrafish lines, which are still under development and not yet widely standardized for immuno-oncology applications.

Second, temperature differences between optimal zebrafish maintenance (28.5 °C) and the conditions required for human cell viability (~34 °C) can affect drug metabolism, immune cell behavior, and overall xenograft performance. Although zebrafish larvae tolerate short-term elevation to 34 °C with high survival rates, careful validation of pharmacokinetics and pharmacodynamics (PK/PD) is necessary to ensure translational relevance [112].

Third, zebrafish complement pathways, while conserved, display distinct kinetics and regulatory features that may influence outcomes of complement-targeted immunotherapies [109]. In addition, species-specific differences in immune receptor homology, tumor signaling networks, and xenobiotic metabolism can result in variations in drug efficacy and toxicity when compared with human systems. These limitations highlight the importance of interpreting zebrafish results within their experimental context and of validating key findings in mammalian models before clinical translation.

A further challenge is the current lack of fully validated tools to assess human-specific immunotherapies. Checkpoint inhibitors and CAR-based cell therapies depend on complex interactions within the human immune system that are not fully recapitulated in standard zebrafish lines. However, recent progress in generating immunodeficient and partially humanized zebrafish strains has enabled the co-engraftment of human immune and tumor cells in vivo. While still in refinement, these models hold strong promise for expanding zebrafish utility in translational immunotherapy testing [107].

Moreover, current zAvatar studies are typically limited to small patient cohorts (<20 cases), which reduces statistical power and generalizability. Larger prospective trials are necessary to validate predictive accuracy across diverse tumor types and treatment modalities [68,69,112].

By addressing these challenges, the zebrafish model can provide a cost-effective, high-throughput, and biologically relevant system for preclinical immuno-oncology research. Its capacity for real-time imaging, genetic manipulation, and rapid drug screening positions zebrafish as a pivotal component in the translational pipeline—one that can accelerate identification of efficacious immunotherapeutic strategies, minimize reliance on resource-intensive rodent models, and, ultimately, expedite the delivery of effective cancer treatments to patients.

Emerging hybrid approaches—such as organoid–zebrafish xenografts, where patient-derived 3D tumor organoids are implanted into zebrafish larvae—are beginning to bridge the gap between organotypic tumor architecture and in vivo drug responses, offering a more physiologically accurate model for personalized therapy assessment. Additionally, AI-assisted image analysis pipelines, using convolutional neural networks, have been successfully applied to quantify tumor growth and invasion in zebrafish glioblastoma models, enabling automated, scalable, and predictive drug-response workflows [113]. These advancements underscore the zebrafish model’s growing potential to seamlessly integrate into translational immuno-oncology, ultimately accelerating the journey from bench to bedside with precision and efficiency.

As translational oncology moves toward functional precision medicine, zebrafish models offer a unique opportunity to integrate biological complexity with clinical urgency. The zAvatar platform enables rapid drug screening on live patient-derived tumor cells, providing actionable results within days. This can directly inform treatment selection in N-of-1 cases, complement genomic data, and guide discussions in molecular tumor boards [103]. In parallel, zebrafish are being used to investigate mechanisms of resistance, immune evasion, and drug synergies—generating insights that feed back into clinical trial design [52]. With ongoing prospective validation in clinical cohorts, zebrafish are positioned not only as a discovery model but also as a decision-support tool in precision oncology, especially in contexts where time, tissue, and resources are limited.

To advance the zebrafish model toward clinical translation, future efforts should focus on standardizing partially humanized zebrafish lines that can support functional human T and B lymphocytes. The development of transgenic lines with fluorescent reporters for adaptive immune subsets would also enable real-time tracking of tumor–immune interactions in vivo. Integrating patient-derived organoids with zebrafish xenografts offers a promising platform to study 3D tumor architecture under dynamic immune pressure. Additionally, coupling zAvatar drug screening with single-cell and spatial transcriptomics may enhance biomarker discovery and therapeutic stratification. Expanding the use of automated AI-driven imaging analysis across different tumor types will also improve scalability and reproducibility, facilitating clinical adoption in precision oncology workflows.

## 8. Conclusions

While zebrafish offer unique advantages for real-time imaging, scalability, and genetic manipulation, certain limitations persist. Physiological characteristics—such as ectothermy and partial divergence in immune regulatory networks—may affect the translational fidelity of some findings. However, recent advances in humanized zebrafish lines, pro-inflammatory cytokine engineering, and improved immune system modeling are helping to bridge these gaps. Importantly, the integration of zebrafish with functional precision oncology approaches, including patient-derived xenografts (zAvatars) and organoid-based transplantation, enables individualized drug testing and therapeutic profiling.

Zebrafish models allow for rapid, high-throughput screening. Their ability to visualize tumor–immune interactions in real time makes them a powerful complement to genomic data. In particular, zAvatars can provide actionable results within days—informing treatment decisions in N-of-1 cases and supporting clinical discussions in molecular tumor boards. Beyond drug selection, zebrafish are increasingly used to explore mechanisms of resistance, immune evasion, and combinatorial strategies, feeding back into the rational design of clinical trials.

With prospective validation underway and expanding methodological refinement, zebrafish are no longer confined to preclinical discovery. They are emerging as a translational tool capable of accelerating the path from functional insight to patient impact—particularly in clinical contexts where time, tissue, and therapeutic clarity are limited.

## Figures and Tables

**Figure 1 jpm-15-00304-f001:**
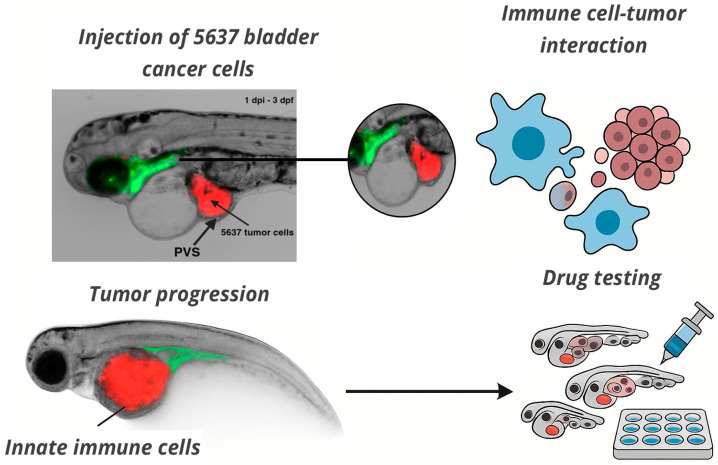
In vivo imaging of tumor–immune interactions in a zebrafish xenograft model. Fluorescently labeled 5637 bladder cancer cells (red, CM-DiI) were microinjected into the perivitelline space (PVS) of 2 days post-fertilization (dpf) zebrafish larvae. The image shows the tumor at 1 day post-injection (1 dpi), when larvae were 3 dpf. Tumor localization and interaction with innate immune cells (green) are visible in real time. This model enables live analysis of tumor progression, immune cell recruitment, and drug response. Imaging was acquired at the INFABiC Facility (Unicamp, Brazil). The schematic illustrates functional applications such as immune–tumor interaction analysis and high-throughput drug screening.

**Figure 2 jpm-15-00304-f002:**
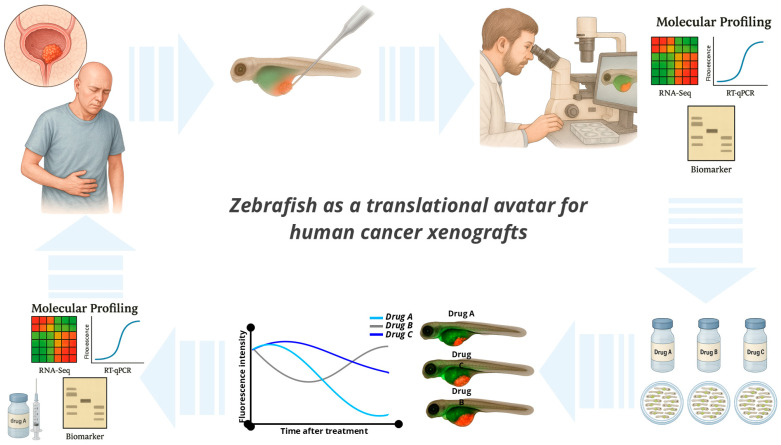
Zebrafish as a translational avatar for human cancer xenografts. Illustration of the zebrafish larval xenograft platform as a versatile in vivo model for cancer research. Fluorescently labeled human tumor cells are microinjected into transparent zebrafish larvae at 2–3 days post-fertilization, enabling real-time visualization of tumor behavior, including proliferation, angiogenesis, invasion, and immune cell interactions. The model supports rapid, high-throughput drug screening via compound immersion and allows for phenotypic readouts within a few days. Combined with genetic tools and imaging capabilities, zebrafish xenografts provide a powerful and scalable platform for translational oncology and functional precision medicine.

**Table 1 jpm-15-00304-t001:** Zebrafish models of solid tumors: transgenic and xenograft approaches.

Tumor Type	Model Type	Genetic Driver/Cell Line	Method/ Injection Site	Key Findings
**Melanoma**	Transgenic	*mitfa:BRAFV600E*, *p53^−/−^*	Stable transgenic line	Mimics melanoma development: cooperation between *BRAFV600E* and *p53* loss induces aggressive tumors [56].
**Melanoma**	Xenograft	A375 (human melanoma)	Perivitelline space (48 hpf)	Supports real-time tracking of invasion and response to BRAF/MEK inhibition [16].
**Liver Cancer**	Transgenic	*fabp10:KRASG12V*	Liver-specific expression	Induces liver tumorigenesis and response to MEK inhibitors [59].
**Colorectal Cancer**	Xenograft	Patient-derived tumor cells	Perivitelline space (48 hpf)	Enabled single-cell analysis of tumor behavior and prediction of chemotherapy responses in vivo [22].
**Bladder** **Cancer**	Xenograft	UM-UC-3; patient-derived tumor fragments	Perivitelline space (48 hpf)	Co-injection of viable BCG with tumor cells; significant tumor regression; the model accurately predicts patient responses to BCG therapy [60].
**Breast** **Cancer**	Xenograft	MDA-MB-231	Automated microinjection into Duct of Cuvier or Perivitelline space	High-throughput xenografts; models tumor growth, metastasis, and drug responses; improves reproducibility for preclinical drug screening [61].
**Glioblastoma**	Xenograft	U87 patient-derived cells	Brain ventricle or yolk sac	Models infiltration and therapy resistance; useful for drug screening [7].

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
