# Peer review of "Zebrafish as a Model for Translational Immuno-Oncology"

_jpm, 2025, doi:10.3390/jpm15070304_

Round 1

Reviewer 1 Report

Comments and Suggestions for Authors

The review emphasizes the application of zebrafish (Danio rerio) as an innovative and alternative preclinical model to investigate immuno-oncology and cancer treatment in a personalized and tailored way. It points to the potential of zebrafish in filling the gap between fundamental in vitro systems and the more costly and longer-lasting mammalian models. Much emphasis is given to embryos being clear, their genetic similarity to humans, and on creating so-called zAvatars (disease-derived xenografts in zebrafish) that allow the rapid testing of personalized treatment approaches, including immunotherapies. The paper covers both the anatomy and physiology of the immune system of zebrafish, tumor models, the possibility of high-throughput pharmacological screening, and the possibility of preclinical validation of novel immunotherapies, including CAR-T and new checkpoint inhibitors. There are also talks about the flaws of the model, such as differences in physiology and the wrong way in which adaptive immunity matures in an embryo.

This study is important because it gives a clear and current perspective of how zebrafish are utilized in modern translational immuno-oncology, with a focus on how the model can be used to develop and verify individualized cancer therapies. The ability to rapidly predict clinical outcomes through zAvatars and the use of zebrafish in high-throughput approaches can shorten the time required for moving research observations from the lab environment to the clinic, while at the same time improving approaches associated with personalized medicine. The focus on innovative approaches, such as combination therapies and genetic editing, makes the article timely and specifically relevant to the modern-day issues encountered in oncology.

It is my belief that the review needs some revisions.

1.While the article is content-heavy, the structure can be enhanced by more delineation of thematic sections, in addition to more use of headings or subheadings. Such an enhancement will guarantee simplicity of comprehension on the part of the reader, taking into consideration the integration of technical details on the immune system, cancer models, and immunotherapy applications. For its importance, the zAvatars section could better be a separate subsection earlier in the book, making it clear from the start the translational research real-world applications of zebrafish.

2. The case for the benefits of using zebrafish is marked by a considerable redundancy, particularly with respect to such aspects as embryonic transparency, high reproductive rate, and genomic homology with humans, being reiterated at intervals. A more concise and combined mention of these features in the introductory section or in a dedicated section can make the text shorter and prevent redundant repetition.

3. In certain instances, assertions are presented, such as "over 85% prediction of clinical response by zAvatars in lung cancer," yet they often lack adequate detail regarding the protocols or clinical conditions under which these studies were conducted. An increase in transparency concerning the sample size, research methodology, and statistical analysis would significantly bolster the credibility of the findings.

4. The manuscript provides a very detailed examination of the zebrafish's capabilities and strengths, which, in view of its function as a review, is entirely acceptable; however, the limitations are only chiefly given near the conclusion. Including reminders of these limitations (e.g., immune maturation, temperature variability, and pharmacokinetic variability) within each of the discrete sections would lead to a more balanced impression of the model per particular use, thereby providing a more complete picture.

5. The writing quality in general is very good and technically correct; however, there are sections with extremely long sentence structures, which could be comprehension-inhibiting, particularly for non-specialist readers. Splitting very long sentences and simplifying the language, wherever feasible, would enhance readability and make the article accessible to a greater number of scientific readers.

6. The part about future prospects gives general advice, like making human zebrafish lines and combining organoid-zebrafish platforms, but it doesn't say what specific methodological goals it wants to achieve. It might be better to be more specific about possible experiments so that the chances are clearer and easier to act on.

Comments on the Quality of English Language

The writing quality in general is very good and technically correct; however, there are sections with extremely long sentence structures, which could be comprehension-inhibiting, particularly for non-specialist readers.

Author Response

Reviewer 1:

Comments and Suggestions for Authors

The review emphasizes the application of zebrafish (Danio rerio) as an innovative and alternative preclinical model to investigate immuno-oncology and cancer treatment in a personalized and tailored way. It points to the potential of zebrafish in filling the gap between fundamental in vitro systems and the more costly and longer-lasting mammalian models. Much emphasis is given to embryos being clear, their genetic similarity to humans, and on creating so-called zAvatars (disease-derived xenografts in zebrafish) that allow the rapid testing of personalized treatment approaches, including immunotherapies. The paper covers both the anatomy and physiology of the immune system of zebrafish, tumor models, the possibility of high-throughput pharmacological screening, and the possibility of preclinical validation of novel immunotherapies, including CAR-T and new checkpoint inhibitors. There are also talks about the flaws of the model, such as differences in physiology and the wrong way in which adaptive immunity matures in an embryo.

This study is important because it gives a clear and current perspective of how zebrafish are utilized in modern translational immuno-oncology, with a focus on how the model can be used to develop and verify individualized cancer therapies. The ability to rapidly predict clinical outcomes through zAvatars and the use of zebrafish in high-throughput approaches can shorten the time required for moving research observations from the lab environment to the clinic, while at the same time improving approaches associated with personalized medicine. The focus on innovative approaches, such as combination therapies and genetic editing, makes the article timely and specifically relevant to the modern-day issues encountered in oncology.

It is my belief that the review needs some revisions.

1.While the article is content-heavy, the structure can be enhanced by more delineation of thematic sections, in addition to more use of headings or subheadings. Such an enhancement will guarantee simplicity of comprehension on the part of the reader, taking into consideration the integration of technical details on the immune system, cancer models, and immunotherapy applications. For its importance, the zAvatars section could better be a separate subsection earlier in the book, making it clear from the start the translational research real-world applications of zebrafish.

A: We thank the reviewer for this insightful suggestion. In the revised manuscript, we implemented a more clearly delineated structure with expanded use of thematic headings and subheadings to improve clarity and reader navigation. We have moved the discussion of patient-derived xenografts (zAvatars) to a standalone section, positioned earlier in the manuscript. This reorganization aims to better highlight the translational and clinical relevance of zebrafish in functional precision oncology, as recommended. We believe this adjustment enhances the overall structure and aligns with the review’s goal of balancing mechanistic insight with clinical applicability.

  1. The case for the benefits of using zebrafish is marked by a considerable redundancy, particularly with respect to such aspects as embryonic transparency, high reproductive rate, and genomic homology with humans, being reiterated at intervals. A more concise and combined mention of these features in the introductory section or in a dedicated section can make the text shorter and prevent redundant repetition.

A: We thank the reviewer for pointing out this redundancy. In the revised manuscript, we consolidated the description of zebrafish-specific advantages (e.g., optical transparency, fecundity, genomic homology) primarily within Section 2, where these features are fully contextualized. The introductory paragraph has been revised to reference these strengths more concisely without reiterating specific details. Similarly, redundant mentions in the Limitations section (Section 7) have been removed or reworded to avoid repetition while maintaining clarity and emphasis on translational relevance. We believe these changes improve the overall focus and readability of the manuscript.

  1. In certain instances, assertions are presented, such as "over 85% prediction of clinical response by zAvatars in lung cancer," yet they often lack adequate detail regarding the protocols or clinical conditions under which these studies were conducted. An increase in transparency concerning the sample size, research methodology, and statistical analysis would significantly bolster the credibility of the findings.

A: We appreciate this valuable observation. To improve transparency, we have now made sure to include more detail on study design, including sample sizes, duration of exposure, and concordance rates between zAvatar predictions and clinical outcomes. These data have been incorporated in Section 6.2, with references to key studies in NSCLC (n=21; 76.9% concordance in 3 days), breast cancer (n=18; 100% match in 10 days), and colorectal cancer (n=20; 95% accuracy). We trust this clarification enhances the credibility of our translational claims.

  1. The manuscript provides a very detailed examination of the zebrafish's capabilities and strengths, which, in view of its function as a review, is entirely acceptable; however, the limitations are only chiefly given near the conclusion. Including reminders of these limitations (e.g., immune maturation, temperature variability, and pharmacokinetic variability) within each of the discrete sections would lead to a more balanced impression of the model per particular use, thereby providing a more complete picture.

A: We thank the reviewer for this important suggestion. In response, we have revised the manuscript to distribute key limitations—such as adaptive immune immaturity, temperature constraints, and pharmacokinetic variability—within the relevant experimental sections (e.g., 4.1.1, 5.2.2, and 6.1). These contextual reminders now provide a more balanced and realistic picture of the zebrafish model’s capabilities and boundaries. Additionally, the main “Limitations and Future Perspectives” section (Section 7) was retained and refined to offer a consolidated view. We believe this integrated approach addresses the reviewer’s concern and enhances the manuscript’s clarity and scientific rigor.

  1. The writing quality in general is very good and technically correct; however, there are sections with extremely long sentence structures, which could be comprehension-inhibiting, particularly for non-specialist readers. Splitting very long sentences and simplifying the language, wherever feasible, would enhance readability and make the article accessible to a greater number of scientific readers.

A: We thank the reviewer for this helpful suggestion. We carefully revised the manuscript to split long and complex sentences, particularly in the Introduction, Sections 3–5, and the Conclusion. We also simplified technical terms where possible to improve readability without compromising scientific rigor. These changes are intended to enhance clarity and accessibility for a broader scientific audience.

  1. The part about future prospects gives general advice, like making human zebrafish lines and combining organoid-zebrafish platforms, but it doesn't say what specific methodological goals it wants to achieve. It might be better to be more specific about possible experiments so that the chances are clearer and easier to act on.

A: We thank the reviewer for this valuable suggestion. In the revised manuscript, we expanded the “Limitations and Future Perspectives” section to include clear methodological directions and concrete experimental strategies. These additions highlight current and emerging priorities in the field, to provide a clearer picture of how the zebrafish model can be methodologically advanced to support translational immuno-oncology and precision medicine.

Comments on the Quality of English Language

The writing quality in general is very good and technically correct; however, there are sections with extremely long sentence structures, which could be comprehension-inhibiting, particularly for non-specialist readers.

A: We appreciate the reviewer’s positive evaluation of the overall writing quality and thank them for pointing out the issue of overly long sentences. In response, we carefully revised the manuscript to improve clarity and readability, particularly for non-specialist readers. We believe these changes have significantly improved the accessibility of the text without compromising scientific rigor.

Reviewer 2 Report

Comments and Suggestions for Authors

The manuscript titled “Zebrafish as a Model for Translational Immuno-Oncology” is well written and presented.

This review highlights the development of the zebrafish immune system and the tools available to track both innate and adaptive immune responses, emphasizing their application in modeling immune evasion, checkpoint molecule expression, and tumor microenvironment dynamics through transgenic and xenograft approaches.

This manuscript is recommended for acceptance after a minor revision to improve clarity and incorporate specific feedback:

  1. In this study, one major limitation is the temperature incompatibility, as zebrafish are maintained at lower temperatures (~28°C), which may not support optimal growth or activity of human cells in xenograft experiments.
  2. There is larval zebrafish possess an immature adaptive immune system, restricting their utility in studying complex T and B cell interactions or long-term immune memory responses.
  3. Interspecies differences in immune signaling pathways, tumor biology, and pharmacokinetics can lead to variations in drug responses compared to humans.
  4. There is a lack of fully validated tools for assessing human-specific immunotherapies, such as checkpoint inhibitors and CAR-based cell therapies, although efforts like humanized zebrafish models are underway to address this.
  5. The authors are encouraged to elaborate more details on the clinical significance and broader implications of this study.
  6. Few sentences could be rephrased for better clarity and readability.

Line 58: “…across developmental stages”.

Line 58: “zAvatars (zebrafish larvae with patient-derived xenografts) enable functional precision oncology”

Line 68: “translational model for immuno-oncology”

Line 122: Similar to humans, zebrafish possess both innate and adaptive immune systems...

Line 133: "In zebrafish and other teleost fish, the head kidney serves as a primary hematopoietic and immune organ..."

Line 158: activation [41]
humans [43]

Line 170: Zebrafish possess professional antigen-presenting cells (APCs), including macrophages and dendritic cells, along with B and T lymphocytes...

Line 174: ...tumor immunity research and immunotherapy studies [26].

Line 533: “ectothermic physiology”

Line 541: Combine or avoid repetition. If “PDX” is used, define it earlier.

Author Response

Reviewer 2:

Comments and Suggestions for Authors

The manuscript titled “Zebrafish as a Model for Translational Immuno-Oncology” is well written and presented.

This review highlights the development of the zebrafish immune system and the tools available to track both innate and adaptive immune responses, emphasizing their application in modeling immune evasion, checkpoint molecule expression, and tumor microenvironment dynamics through transgenic and xenograft approaches.
A: We sincerely thank the reviewer for the positive evaluation of our manuscript.

This manuscript is recommended for acceptance after a minor revision to improve clarity and incorporate specific feedback:

  1. In this study, one major limitation is the temperature incompatibility, as zebrafish are maintained at lower temperatures (~28°C), which may not support optimal growth or activity of human cells in xenograft experiments.

A: We appreciate the reviewer’s observation. While the standard zebrafish maintenance temperature is 28.5 °C, xenotransplantation protocols in larvae commonly adopt elevated incubation temperatures between 32 °C and 35 °C to enhance the viability and proliferation of human cells. These conditions are generally well tolerated by zebrafish embryos.

Although the optimal balance between temperature and incubation duration should be determined empirically for each tumor model, short-term incubation at 34 °C has consistently proven to be safe and effective for rapid functional testing. We have now included a brief clarification regarding this methodological aspect in the revised manuscript, along with appropriate references

2. There is larval zebrafish possess an immature adaptive immune system, restricting their utility in studying complex T and B cell interactions or long-term immune memory responses.

A: We thank the reviewer for pointing out this critical limitation. Indeed, larval zebrafish rely predominantly on innate immunity during early development, with full maturation of adaptive responses—including functional T and B cell repertoires—occurring several weeks post-fertilization. Consequently, while larval models are highly suited for investigating innate immune mechanisms and rapid tumor–immune interactions, they are limited in studies requiring long-term immune memory or antigen-specific T/B cell dynamics. We have now added a brief clarification on this limitation in the corresponding section.

3. Interspecies differences in immune signaling pathways, tumor biology, and pharmacokinetics can lead to variations in drug responses compared to humans.

A: We thank the reviewer for raising this important point. As with any non-mammalian model, zebrafish do exhibit interspecies differences in immune regulation, tumor signaling dynamics, and drug metabolism when compared to humans. These differences may influence drug response outcomes and limit the extrapolation of certain mechanistic findings. Nevertheless, zebrafish offer high-content, high-throughput platforms for early-stage screening, functional phenotyping, and comparative analysis. When interpreted within their translational scope—and complemented by mammalian validation—zebrafish data provide valuable insights that can guide downstream preclinical and clinical decision-making. We have added a corresponding statement in the limitations section of the revised manuscript.

4. There is a lack of fully validated tools for assessing human-specific immunotherapies, such as checkpoint inhibitors and CAR-based cell therapies, although efforts like humanized zebrafish models are underway to address this.

A: Indeed, zebrafish models currently face limitations in fully replicating human-specific immunotherapy mechanisms. While these models are still under validation and not yet widely available, they represent a promising direction toward enabling functional assessment of human-targeted immunotherapies. We have added a brief discussion of this limitation and future outlook in the revised limitations section.

5. The authors are encouraged to elaborate more details on the clinical significance and broader implications of this study.

A: We appreciate the reviewer’s suggestion to strengthen the translational perspective. In response, we expanded the final section to explicitly highlight the clinical significance of zebrafish models in precision oncology. We emphasized how patient-derived zebrafish avatars (zAvatars) can contribute to individualized therapeutic decision-making, facilitate functional drug testing in real time, and support prospective trials aiming to validate this approach in clinical practice. This addition underscores the role of zebrafish as a complementary platform in the translational pipeline, bridging experimental biology and patient care.

6. Few sentences could be rephrased for better clarity and readability.

A: Thank you for your valuable comment. We have carefully corrected the sentences according to your suggestions to enhance clarity and readability.

Line 58: “…across developmental stages”.

Line 58: “zAvatars (zebrafish larvae with patient-derived xenografts) enable functional precision oncology”

Line 68: “translational model for immuno-oncology”

Line 122: Similar to humans, zebrafish possess both innate and adaptive immune systems...

Line 133: "In zebrafish and other teleost fish, the head kidney serves as a primary hematopoietic and immune organ..."

Line 158: activation [41]
humans [43]

Line 170: Zebrafish possess professional antigen-presenting cells (APCs), including macrophages and dendritic cells, along with B and T lymphocytes...

Line 174: ...tumor immunity research and immunotherapy studies [26].

Line 533: “ectothermic physiology”

Line 541: Combine or avoid repetition. If “PDX” is used, define it earlier.

A: We have revised and corrected the sentences accordingly

Round 2

Reviewer 1 Report

Comments and Suggestions for Authors

The authors have revised their manuscript in accordance with the suggestions.